# A Flexible Magnetic Field Sensor Based on PZT/CFO Bilayer via van der Waals Oxide Heteroepitaxy

**DOI:** 10.3390/s23229147

**Published:** 2023-11-13

**Authors:** Weijuan Pan, Yuan Ao, Peng Zhou, Leonid Fetisov, Yuri Fetisov, Tianjin Zhang, Yajun Qi

**Affiliations:** 1Ministry of Education Key Laboratory for Green Preparation and Application of Functional Materials, Hubei Provincial Key Laboratory of Polymers, Collaborative Innovation Center for Advanced Organic Chemical Materials Co-Constructed by the Province and Ministry, School of Materials Science and Engineering, Hubei University, Wuhan 430062, China; 202121113012808@stu.hubu.edu.cn (W.P.); 202031113010091@stu.hubu.edu.cn (Y.A.); zhou@hubu.edu.cn (P.Z.); zhangtj@hubu.edu.cn (T.Z.); 2Research-Education Center “Magnetoelectric Materials and Devices”, MIREA—Russian Technological University, Moscow 119454, Russia; fetisovl@yandex.ru (L.F.); fetisov@mirea.ru (Y.F.)

**Keywords:** ME coupling, ME magnetic field sensors, flexible, clamping effect

## Abstract

Magnetoelectric (ME) magnetic field sensors utilize ME effects in ferroelectric ferromagnetic layered heterostructures to convert magnetic signals into electrical signals. However, the substrate clamping effect greatly limits the design and fabrication of ME composites with high ME coefficients. To reduce the clamping effect and improve the ME response, a flexible ME sensor based on PbZr_0.2_Ti_0.8_O_3_ (PZT)/CoFe_2_O_4_ (CFO) ME bilayered heterostructure was deposited on mica substrates via van der Waals oxide heteroepitaxy. A saturated magnetization of 114.5 emu/cm^3^ was observed in the bilayers. The flexible sensor exhibited a strong ME coefficient of 6.12 V/cm·Oe. The local ME coupling has been confirmed by the evolution of the ferroelectric domain under applied magnetic fields. The flexible ME sensor possessed a stable response with high sensitivity to both AC and DC weak magnetic fields. A high linearity of 0.9988 and sensitivity of 72.65 mV/Oe of the ME sensor were obtained under flat states. The ME output and limit-of-detection under different bending states showed an inferior trend as the bending radius increased. A flexible proximity sensor has been demonstrated, indicating a promising avenue for wearable device applications and significantly broadening the potential application of the flexible ME magnetic field sensors.

## 1. Introduction

ME coupling materials have garnered extensive research attention due to their unique coupling effect between magnetic and electrical sequences. The nano-devices fabricated by the nanocomposite film of ferroelectric and ferromagnetic materials have achieved high coupling coefficients. This has ignited technological advancements, particularly in the context of their potential applications in next-generation magnetic sensors, such as soft robotics, biomedicine and diagnostics of disease, and consumer electronics [1,2,3,4,5,6,7]. In contrast to flux sensors, such as a superconducting quantum interference device magnetometer (SQUID), an atomic magnetometer (AMM), and flux-gate meters (FGM), ME sensors are inherently field-sensitive, rendering them independent of device size [8,9]. This characteristic is advantageous for the development of miniaturized devices. To improve ME detection performance, various ferroelectric and ferromagnet structures have been developed to improve ME response and strain transfer. Ferroelectric/ferromagnetic composites of types 1-3, 0-3, and 2-2 have been fabricated with various methods [10,11]. The 2-2 type composites are particularly effective in mitigating leakage current generation [10]. The tightly bonded layer structure facilitates strain transfer, consequently leading to significantly larger ME coefficients [12]. Nevertheless, due to the clamping effect, the ME coupling response is still inferior to the expected value [13,14]. Transferring oxide films from rigid substrates to some flexible substrates is a common method to release the clamping effect. Kim et al. transferred BaTiO_3_/CoFe_2_O_4_(CFO) oxide films from MgO substrates to polydimethylsiloxane by etching MgO. Compared to the film deposited on MgO, flexible BTO-CFO films exhibited outstanding mechanical properties and enhanced their integration into microelectromechanical systems [15]. Moreover, it is worth noting that oxide films can be released using Sr_3_Al_2_O_6_, a supercritical layer, which is a prevalent technique for fabricated flexible films [16]. Preparing ME films on flexible substrates is a more efficient and convenient method to enhance the magnetoelectric coupling coefficient. Meanwhile, flexible electronic components have gained widespread interest with the high-speed evolution of the Internet of Things and implantable technology. The manufacturing of wearable flexible devices aligns well with current trends, as flexible electronic devices enable electronic devices to satisfy a broader range of application demands [17,18,19,20].

As a novel type of flexible substrate, mica has strong intra-layer interactions and frail inter-layer interactions, which enable ferroelectric oxides to be deposited via van der Waals epitaxy [21,22,23,24]. Simultaneously, mica substrates offer a range of unique advantages, such as excellent thermal conductivity, atomically smooth surfaces, chemical inertness, and good mechanical flexibility at thicknesses below 100 microns [24,25,26,27]. Furthermore, the high melting point of mica (1150–1300 K) can be compatible with most modern film preparation processes, which isconducive to the deposition of oxide films at high temperatures [22,23,24]. The interface between a mica substrate and oxide film is mainly mediated by van der Waals forces. The weak van der Waals force is 1–4 orders of magnitude weaker than ionic or covalent bonds, thus significantly reducing the clamping effect [28]. Extensive research has indicated that high quality oxide epitaxial films can be fabricated on mica substrates, and the many features of mica make it a candidate for the fabrication of flexible devices. Chu’s group epitaxially grew self-assembled BFO-CFO heterojunctions on mica substrates. Owing to the weak interaction between the muscovite and the deposited heterojunction, the clamping effect was reduced, and a phase-field simulation model was used to supplement the explanation. The ME coefficient of this heterojunction is 74 mV/cm·Oe, which is greater than the ME coefficient previously reported on flexible substrates [29]. Utama et al. fabricated ZnO nanowires directly on mica substrates via van der Waals epitaxy without any seed layers. An extensive analysis, including with electron microscopy, showed that well-crystallized epitaxial nanowires on muscovite did not require lattice matching. Their results demonstrate that van der Waals epitaxy, as a universal epitaxy strategy, can be applied to a broad spectrum of materials and epitaxial nanostructures [30].

In this work, a flexible magnetic field sensor based on the PbZr_0.2_Ti_0.8_O_3_ (PZT)/CFO bilayer deposited on mica by van der Waals interaction was proposed. The weak van der Waals force between mica substrates and oxide films significantly reduced the substrate clamping effect. Meanwhile, the layered structure has a larger contact area, which is conducive to stress transfer. These properties are expected to allow for the structure to exhibit an enhanced ME response. We characterized the crystal structure of the sample using X-ray diffraction to verify the epitaxial growth of the sample and examined the magnetic and electrical properties of the sample using vibrating sample magnetometer (VSM) and piezoresponse force microscopy (PFM), respectively. To confirm the occurrence of ME responses, we conducted experiments that involved monitoring changes in the dielectric constant and dielectric loss under various magnetic fields as well as observing the switching of ferroelectric domains. The effect of strain on the detection performance of the sensor was investigated by changing the bending state of the sample and examining the ME voltage output and magnetic field limit-of-detection. Finally, the practical potential of this flexible ME sensor was also evaluated.

## 2. Materials and Methods

The flexible ME films with SrRuO_3_ (SRO) electrodes buffered with NiFe_2_O_4_ (NFO) were grown on a mica substrate using pulsed laser deposition with a KrF (λ = 248 nm). The size of the mica substrate was 5 mm × 5 mm, and the thickness was 200 μm. All films were deposited at a frequency of 5 Hz. CFO was deposited on SRO electrodes at 730 °C with an oxygen partial pressure of 15 Pa. PZT was deposited at 650 °C at an oxygen partial pressure of 5 Pa. The samples were then annealed at 650 °C under 500 Pa oxygen pressure for 30 min at the end of the PZT deposition process, further cooled to 400 °C at a cooling rate of 10 °C min^−1^, and then allowed to cool naturally to room temperature. We used a Pt top electrode with a diameter of 200 μm on the surface of PZT with magnetron sputtering.

The structural properties of the layered film were measured with Bruker D8 Discover diffraction system (CuKα1, λ = 1.5406 Å). The ferroelectric domain switch of different magnetic fields was investigated with PFM using an Asylum Research MFP-3D atomic force microscope (AFM). The *P*–*E* loop was obtained with macroscopic ferroelectric properties tests on the sample using the ferroelectric material test system (Radiant Premier II, Radiant Technologies Inc., Albuquerque, NM, USA) with a testing frequency of 100 Hz. The diverse stress states were obtained by fixing the film on different bending molds. The ME coupling signals of flexible heterojunction evoked by oscillating magnetic field produced by a standard AC current source meter (Keithley 6221, Beaverton, OR, USA) were examined under various DC magnetic fields with lock-in modulation technology (SR830, Stanford Research Systems Inc., Sunnyvale, CA, USA).

## 3. Results

Figure 1a shows the structural scheme of PZT/CFO bilayer films grown on mica substrates with a thickness of 200 μm. NFO was used as a buffer layer while SRO as the bottom electrode. The XRD pattern of the film in Figure 1b was along the out-of-plane (OOP) direction. A series of diffraction peaks were observed in the XRD *θ*–2*θ* scan pattern (Figure 1b), including mica{001}, SRO{111}, CFO{111}, and PZT{111}. These {111}-type diffraction peaks indicated the epitaxial characteristics of the ME heterojunction and the absence of other impurities, indicating that the film is of high crystal quality. PFM was used to examine the local ferroelectric response to further characterize the quality of the film. Figure 1c shows the surface morphology of PZT/CFO film measured with AFM. The root-mean-square roughness of the PZT/CFO bilayer film surface is ~5.829 nm, indicating the high quality of the film. A box-in-box polarization switch pattern was written on the PZT layer by applying an electric field to the sample through the conductive tip. A tip bias of +20 V was applied to the 12 μm × 12 μm square area, while a tip bias of −20 V was applied to the inner 4 μm × 4 μm square area. The corresponding out-of-plane amplitude information of the film is shown in Figure 1d. When various voltages were applied to polarize the PZT film, the PZT layer underwent deformation along the electric field direction due to the converse piezoelectric effect, resulting in a change in surface amplitude. In Figure 1e, the PFM phase image shows a clear contrast area between light and dark, with the bright area of 12 μm × 12 μm corresponding to upward polarization and 4 μm × 4 μm central region corresponding to downward polarization. The polarization switching followed by the applied electric field indicates the excellent ferroelectric performance of the PZT layer.

Figure 2a shows a typical *P*–*E* loop of PZT/CFO bilayer film at room temperature. According to the ferroelectric hysteresis loop, the remanent polarizations ±*P*_r_ were obtained about 24.6 μC/cm^2^ and −24.7 μC/cm^2^ and the coercive voltage were obtained about 2.27 V and −1.37 V. Well-saturated P–E loop indicated excellent properties of PZT/CFO film. Figure 2b shows the magnetic hysteresis loop of the PZT/CFO bilayer measured at 300 K by the VSM. The film exhibited typical ferromagnetic characteristics, with a saturation magnetization of approximately 114.5 emu/cm^3^ and a coercive field of approximately 1.47 kOe. Figure 2c shows the ME coefficient of the PZT/CFO bilayer film. As the DC magnetic field increases, the ME coefficient shows a trend of decreasing, with a maximum ME coefficient of 6.12 V/cm·Oe at 57 kHz.

The ME coupling effect was also confirmed with the magnetodielectric effect, as shown in Figure 3a. All dielectric constant curves showed a decreasing trend with an increasing magnetic field. The frequency-dependent dielectric constants and losses were recorded under a DC magnetic field of 450 Oe and 800 Oe. The result shows that the dielectric constants decrease with the increased external magnetic field, indicating the occurrence of magnetodielectric (MD) effect. An effective MD constant is defined by the following:(1)$$MD(\%)=\frac{∆\varepsilon_{r}}{\varepsilon_{r}}\times 100$$

Thus, an MD value as high as 20.84% at 10 kHz was obtained, indicating a possible dynamic coupling between magnetization and polarization [31,32]. As the magnetic field increased, the total deformation caused by the magnetostrictive effect further increased and transferred to the PZT layer through the interface, resulting in a lower dielectric constant [33]. Meanwhile, a broad peak is observed in the dielectric loss curves that move towards the low frequency as the magnetic field is increased, as shown in Figure 3b. These features further confirmed the ME coupling in the as-prepared bilayers [34].

The interface is a vital factor that determines the comprehensive performance of composite materials [35,36,37]. In oxide heterostructures, semi-coherent-like interfaces are commonly observed, where bound interface charges, stress, and defects may exist. In the case of layered composite materials, diffusion extends along the entire interface in addition to the grain boundaries. The dielectric constant is affected by the interaction between magnetic and electric dipoles at room temperature as well as the deformation at the interface. The presence of a specific binding interface charge in the bilayer, which refers to chemical mixing at the bilayer interface, could lead to alterations in the dielectric properties between the bilayers. Meanwhile, the oxygen vacancies near the interface can also affect leakage current and magnetism.

To further investigate the micro-magnetoelectric properties, PFM was used to study the distribution of ferroelectric domains in films under different magnetic fields. The morphology, amplitude, and phase images without a magnetic field are shown in Figure 4a–c. Meanwhile, ferroelectric domains showed a strong correlation with amplitude but were independent of surface morphology. The white and purple areas in Figure 4c represented the area with polarization up and down, respectively. The large-scaled domains in Figure 4c demonstrated long range ferroelectric ordering, which is an inherent feature of ferroelectrics. When 6 kOe and −6 kOe in-plane magnetic fields use a variable field module (VFM), the upward polarization region in the red and blue circles decreased significantly while the upward polarization region in the green circle increased significantly, as shown in Figure 4d,e. One may note that a small area of complete contrast changing from white to purple or vice versa can be seen by applying a positive or negative magnetic field. Since CFO is a negative magnetostrictive material, when the PZT/CFO ME heterojunction was subjected to a horizontal magnetic field, the CFO layer will generate compressive stress along the magnetic field direction and transfer it to the PZT layer, thus generating the output voltage via piezoelectric effect. Thus, stress-induced ME coupling in PZT/CFO bilayer films was confirmed by switching oflocal ferroelectric domains under different magnetic fields [38].

Figure 5a shows the ME output voltage of the PZT/CFO film under various *H*_ac_ magnetic fields. When the magnetic field was 75 μT, the ME voltage reached its maximum value of 0.057 mV. Based on the linear fitting from the data shown in Figure 5a, a linearity of 0.9988 and a sensitivity of 72.653 mV/Oe were obtained for the AC magnetic field sensor operating at a resonant frequency of 57 kHz. The linearity showed that the ME sensor has high sensitivity and broad application prospects. By continuously reducing the difference between the two currents to reduce the difference between the magnetic fields, the limit-of-detection of the sensor under DC and AC magnetic fields was ultimately obtained, as shown in Figure 5b. The results showed that the DC field limit-of-detection was 2 mT and the AC field resolution was 5 nT, respectively. Furthermore, Table 1 presents a comparison of the limit of detection among various flexible magnetic field sensors. Portable electronic devices and wearable health monitoring systems have attracted widespread attention due to their flexible and lightweight characteristics, and there is a strong demand to interconnect multiple sensing elements with multiple devices to form a complex platform, further promoting the application of wearable devices for multifunctional purposes. Owing to mica’s inherent flexibility, ME sensors produced on mica substrates have remarkable flexibility, potentially serving as a novel component for wearable devices and thus broadening the application of ME sensors. Functional oxides via vdW epitaxy on mica have been demonstrated for next-generation flexible devices owing to their unique set of features, such as being lightweight, transparent, flexible with an atomically smooth surface, chemically inert, and stable at high temperatures. Specifically, for ME bilayers, due to the weak interaction between mica substrates and the functional films, there is a good opportunity to overcome the substrate clamping effect and enhance the ME coupling response, which will improve the sensing performance of the ME sensors. Based on the flexibility of the mica substrate, we made bending studies on the sensor. The sample was then bent to obtain different stress states by fixing it to molds with different bending radii. Figure 5c shows the ME output with an increasing magnetic field under a series of bending radii. The results showed that the ME output voltage of the sample decreases as the bending radius increases regardless of whether it was in the flex-in or flex-out bending state. Simultaneously, the linearity and sensitivity under different bending radii can be calculated, and the results showed that the change in linearity under different bending radii was very small, but the sensitivity decreased as the bending radius increased, with a minimum of 36.3 mV/Oe, as shown in Table 2. The ME coupling in the as-prepared PZT/CFO heterojunction has been proven as a strain-mediated ME coupling. The additional strain may also contribute to the ME outputs and further tune the detecting properties of the sensors, which provides flexibility and new application routes for the magnetic field sensors.

In general, wearable electronics consist of six major constituents: (i) substrate, (ii) sensors, (iii) actuators, (iv) interconnects, (v) wireless transmission, and (vi) power [17]. All components require high mechanical robustness and operational stability, and each component needs to be capable of withstanding high levels of operational strain. Thus, it is worthwhile to mention that the magnetic field limit-of-detection of PZT/CFO films under different strain conditions is worth studying for various application conditions. The effect of mechanical strain on the magnetic field limit-of-detection was investigated at a resonance frequency of 57 kHz, as shown in Figure 6. Regardless of whether the bending radius increases in the inner or outer bending states, the limit-of-detection showed a decreasing trend. This result was consistent with the variation trend of ME voltage under different stress states, as shown in Figure 5c. In the case of flex-in and flex-out bending with a bending radius of 5 mm, the AC magnetic field limit-of-detection was 30 nT and 40 nT, respectively, as shown in Figure 6a,b. The DC magnetic field limit-of-detection was 4 mT and 3 mT, respectively, as shown in Figure 6c,d. As the bending radius increased, the limit-of-detection did not decrease significantly, indicating that PZT/CFO ME magnetic field sensors based on flexible mica substrates have potential applications.

The slight change of the limit-of-detection may due to extra mechanical strain caused by the flexible mica substrate bending. In general, physical strain S imposed on a *t*_film_-thick film layer by bending a *t*_sub_-thick substrate to a certain radius of curvature r is given by the following:(2)$$\varepsilon =\left(\frac{t_{sub}+t_{film}}{2R}\right)\left[1+\frac{\left(1-\chi \right)\eta }{\left(1+\eta \right)\left(1+\chi \eta \right)}\right],$$
where *η* = *t*_film_/*t*_sub_, *χ* = *Y*_film_/*Y*_sub_, *Y*_film_, and *Y*_sub_ are the Young’s modulus of the film and the substrate, respectively [44]. By applying various strains to the sensor under different bending radii, it couples with the magnetostrictive effect as well as the piezoelectric effect, which results in different ME responses and also the variation of the ME output voltages.

Compared to traditional chemical and tactile sensors, flexible magnetic field sensors as wearable electronic devices have remote motion characteristics and can be driven by surrounding magnetic fields to achieve non-contact skin interaction. A flexible proximity sensor demonstration was presented, as shown in Figure 7a. By fixing the sensor and moving the magnet back and forth within a certain distance range, the output voltages have been recorded, as shown in Figure 7b. One can see that there was a significant change in the ME output voltage as the magnet moved back and forth. The peak value of the ME output voltage represented the proximity of the sensor to the magnet and kept a stable value at about 12 mV. The values have significant repeatability and accuracy at different distances from the sample, indicating the potential application of this flexible heterojunction as a magnetic field sensor.

## 4. Conclusions

In summary, the flexible magnetic field sensor based on a PZT/CFO bilayer deposited on mica via van der Waals heteroepitaxy was constructed and confirmed its promising application as a motion sensor. The XRD patterns unequivocally demonstrate the epitaxial nature of the oxide film on mica substrates. The high crystallinity serves as a crucial assurance for the practical application of ME magnetic field sensors. The *P*–*E* loop and the *M*–*H* loop of the PZT/CFO film confirm the excellent electrical and magnetic properties of the as-prepared films. Magnetic dielectric tests conducted unequivocally confirmed the presence of the ME effect. Specifically, the MD% value of about −20.84% was obtained under 800 Oe at 10 kHz. Under 6 kOe and −6 kOe, the direction of the ferroelectric domain underwent a significant transformation relative to 0 Oe, demonstrating that under the action of an external magnetic field, the strain generated by the ferromagnetic layer was successfully transferred to the ferroelectric layer and caused a change in the polarization direction of the ferroelectric domain. The AC and DC magnetic field limit-of-detection of the ME magnetic sensor in the flat state were 5 nT and 2 mT, respectively, and showed an inferior trend with the decrease in bending radius. Flexible ME sensors for detecting motion were also presented, showing the promise of application prospects and providing a novel approach for manufacturing and integrating health detection and wearable flexible ME devices.

## Figures and Tables

**Figure 1 sensors-23-09147-f001:**
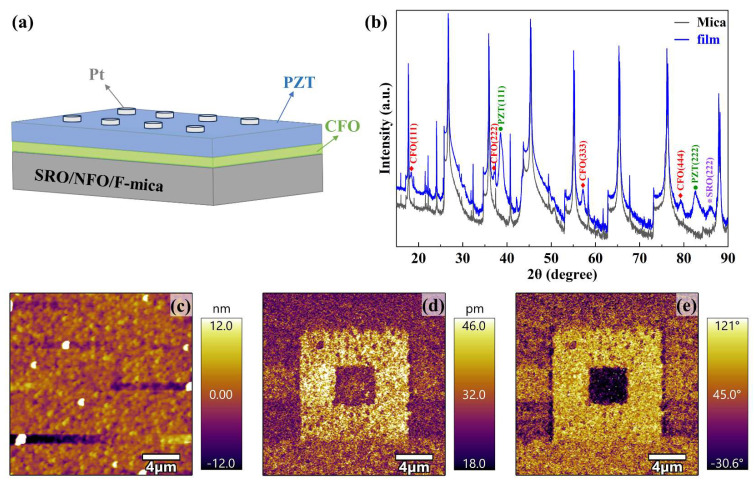
(**a**) Schematic diagram of 2-2 type layered film; (**b**) The XRD *θ*–2*θ* scan of the PZT/CFO film and mica substrate. Characterization of ferroelectric properties of PZT/CFO films via PFM. (**c**) Morphology; (**d**) amplitude; (**e**) phase.

**Figure 2 sensors-23-09147-f002:**
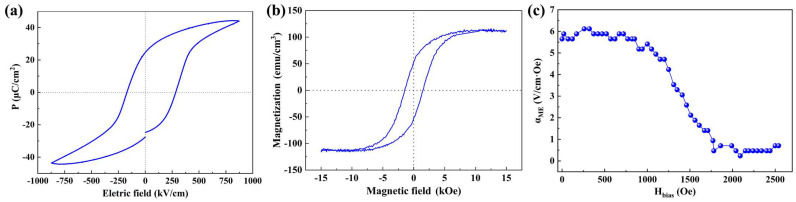
(**a**,**b**) *P*–*E* loop and *M*–*H* loop for PZT/CFO film along OOP direction; (**c**) *α*_ME_ of PZT/CFO bilayer film with increasing DC magnetic field.

**Figure 3 sensors-23-09147-f003:**
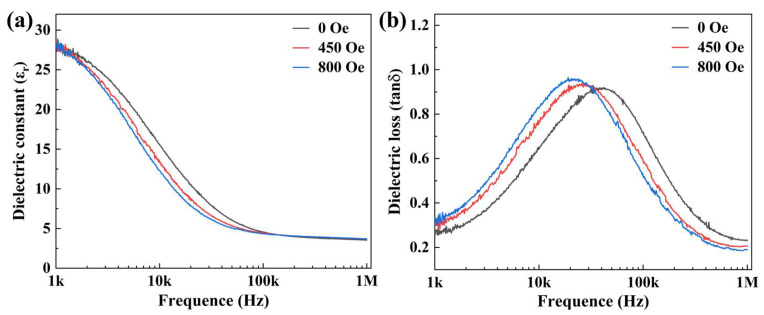
Dielectric performance characterization under different magnetic fields. (**a**) Dielectric constant; (**b**) dielectric loss.

**Figure 4 sensors-23-09147-f004:**
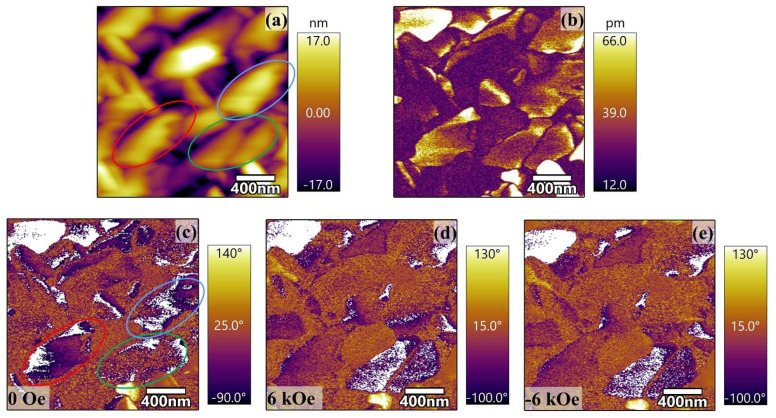
(**a**–**c**) The morphology, PFM amplitude, and phase of PZT/CFO bilayer films under an external magnetic field of 0 Oe; (**d**,**e**) PFM the phase image under a magnetic field of 6 kOe and −6 kOe, respectively.

**Figure 5 sensors-23-09147-f005:**
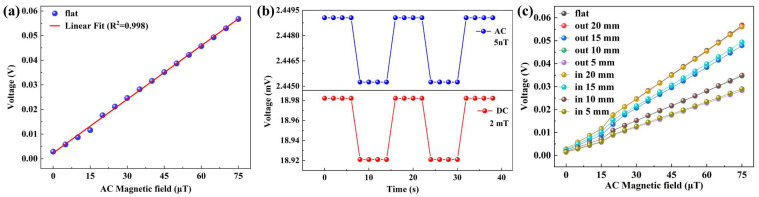
(**a**) ME output voltage of PZT/CFO bilayer film in the range of 0 μT–75 μT; (**b**) AC and DC magnetic field limit-of-detection in the unbending state; (**c**) ME output voltage under different bending states.

**Figure 6 sensors-23-09147-f006:**
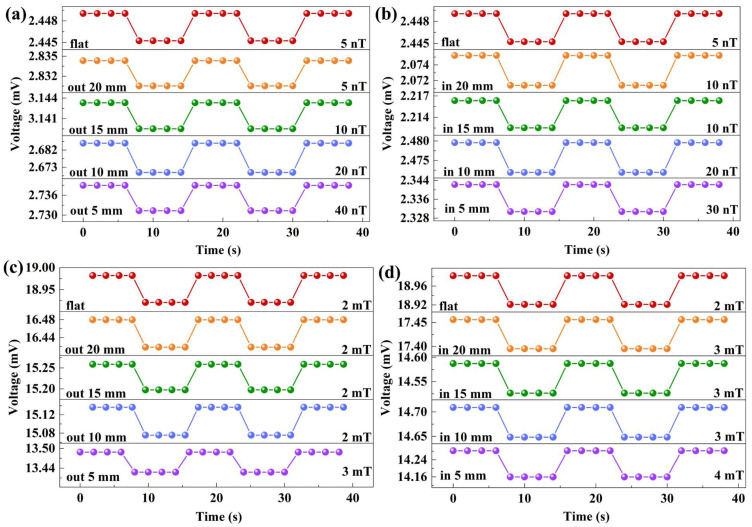
(**a**,**b**) AC magnetic field limit-of-detection under various bending radii; (**c**,**d**) DC magnetic field limit-of-detection under various bending radii.

**Figure 7 sensors-23-09147-f007:**
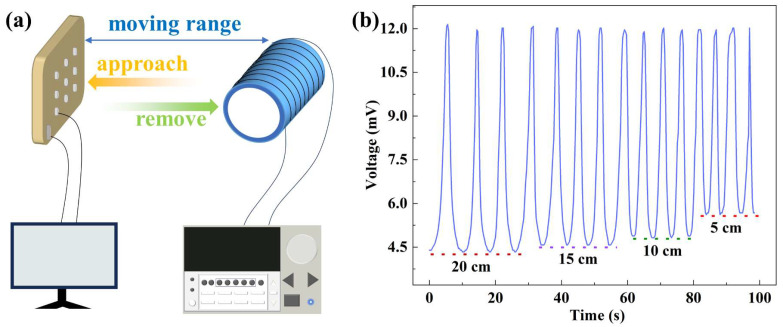
(**a**) Layout schematic diagram of the ME magnetic field sensor; (**b**) voltage output of moving the sample back and forth within a certain range.

**Table 1 sensors-23-09147-t001:** A comparison of flexible magnetic field sensors.

Composition/Materials	AC/DC	Limit of Detection	Ref
Metglas/PVDF	AC	10 pT	[39]
ZnO	AC	120 nT	[40]
Pb(Zr_0.52_Ti_0.48_)O_3_/Metglas	AC	0.2 nT	[1]
Metglas/PVDF/Metglas	AC	8 nT	[41]
Ni_0.81_Fe_0.19_/PR/PET	AC	150 nT	[42]
Pb(Zr_0.20_Ti_0.80_)O_3_/CoFe_2_O_4_	AC	5 nT	This work
AgNWs/PDMS/Fe_3_O4	DC	0.5 mT	[43]
Metglas/PVDF	DC	8 nT	[39]
Metglas/PVDF/Metglas	DC	8 nT	[41]
Pb(Zr_0.20_Ti_0.80_)O_3_/CoFe_2_O_4_	DC	5 mT	This work

**Table 2 sensors-23-09147-t002:** The linearity and sensitivity under different bending radii.

Bending Radius (mm)	Sensitivity (mV/Oe)	Linearity
Flat	72.654	0.9988
Flex-out 20	72.253	0.9986
Flex-out 15	62.125	0.9980
Flex-out 10	44.398	0.9986
Flex-out 5	36.302	0.9987
Flex-in 20	71.990	0.9987
Flex-in 15	63.244	0.9987
Flex-in 10	44.698	0.9985
Flex-in 5	37.031	0.9986

## Data Availability

Data are contained within the article.

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
