# Peer review of "A Flexible Magnetic Field Sensor Based on PZT/CFO Bilayer via van der Waals Oxide Heteroepitaxy"

_sensors, 2023, doi:10.3390/s23229147_

Round 1

Reviewer 1 Report

Comments and Suggestions for Authors

In the manuscript, a flexible magnetic field sensor based on PZT/CFO bilayer deposited on mica via van der Waals heteroepitaxy has been fabricated. The authors provided full characteristics on the quality of the PZT/CFO bilayer thin films including the XRD, P-E loops and the M-H loops as well as the magnetodielectric properties and the ME coupling constants. More importantly, the sensing performance as well as the flexibility of the sensors for both the weak AC and DC magnetic field were also illustrated. A high linearity of 0.9988 and sensitivity of 72.65 mV/Oe of the ME sensor were obtained under flat states for the as-prepared sensors. flexible proximity sensor has been demonstrated, indicating a promising avenue for wearable device applications and significantly broadening the potential application of the flexible ME magnetic field sensors.

The manuscript is well written and discussed. I would like to recommend this paper to publish in Sensors after clear following comments.

1. A comparation of the sensitivity with other reported flexible sensors is needed.

2. The benefit for ME bilayer thin films grown on flexible substrate need further discussion.

3. Page 5, line172, “…while the upward polarization region in the blue circle increased….”here “the blue circle” should be the “the green circle”.

Comments on the Quality of English Language

Minor editing of English language required

Reviewer 2 Report

Comments and Suggestions for Authors

Described in this manuscript, the authors synthesized PZT/CFO films on a flexible substrate. The magnetoelectric performance and strain effect are characterized. The data look reliable and the discussion is generally solid. However, I would like to see the following points addressed in the publishable version.

1. In Fig. 1b, some XRD peaks that are additional to the substrate pattern are not indexed, for example, the peak at around 42 degree. What is the possible source?

2. In the PFM images shown in Fig. 1, why does the domain boundary look notably more intense in the phase image than it looks in the amplitude image?

3. The experimental details of the P-E loop in Fig. 2 are missing. For examples, what the frequency was and what type of measurement equipment was used. Also, better to convert the voltage into the electric field (voltage divides by film thickness). 

4.  What temperature was the magnetodielectric measurement (Fig. 3) performed at? 

5. About Fig. 4, I believe more discussion should be added to explain why magnetic fields in opposite directions flip the polarization in the same way. The authors have done linear fitting of the induced voltage and the magnetic field magnitude, which implies that the observed ME effect is a linear ME effect. However, for a linear ME effect, magnetic fields in opposite directions are supposed to cause opposite polarization changes, which seems not consistent with the result shown in Fig. 4.

6. What causes the anomaly of induced voltage around 15 - 20 uT in Fig. 5c?

7. Could the authors comment on the effect of bending histories? For example, if the film is bended for several times and then released, is the ME performance identical with the pristine film?

End of my report.

Comments on the Quality of English Language

The English in this manuscript is generally good.

Reviewer 3 Report

Comments and Suggestions for Authors

1) The authors state, “...the linearity and sensitivity under different bending radii can be calculated, and the results showed that the change in linearity under different bending radii was very small, but the sensitivity decreased as the bending radius increased, with a minimum of 36.3 mV/Oe...” How are the linearity and sensitivity of the sensor affected by different bending radii?

2) The influence of interdiffusion processes at the interface (between the ferroic phases) is a significant factor in the ME coupling response. However, the authors do not address such effects. Could the insights from the following references, https://doi.org/10.1016/j.materresbull.2023.112169 and https://doi.org/10.1016/B978-0-323-90586-2.00003-6, provide a deeper understanding of this aspect.

3) How does strain-mediated performance impact the ME coupling in the PZT/CFO heterojunction?

4) Based on Table 1, which lists linearity and sensitivity under different bending radii, are there theoretical implications or models that can predict these variations?

5) The magnetic field detection limit under different strain conditions was investigated. Are there other environmental or operational factors besides strain that could affect this limit?

6) How might the observed decrease in sensitivity with increased bending radii be addressed to improve the overall performance and versatility of the PZT/CFO heterojunction sensor in flexible applications?

Round 2

Reviewer 2 Report

Comments and Suggestions for Authors

The authors have addressed all of my concerns. I agree on the publication of the revised manuscript.

Comments on the Quality of English Language

The English of this manuscript is generally good.